# Identifying Urban Park Events through Computer Vision-Assisted Categorization of Publicly-Available Imagery

**Yizhou Tan [1], Wenjing Li [2]**, **Da Chen [3] and Waishan Qiu [4],***

1    Department of Architecture, Rhode Island School of Design, Providence, RI 02903, USA; ytan02@risd.edu
2    Center for Spatial Information Science, The University of Tokyo, Tokyo 277-0882, Japan; liwenjing@csis.u-tokyo.ac.jp
3    Department of Computer Science, The University of Bath, Bath BA2 7AY, UK; da.chen@bath.edu
4    Department of Urban Planning and Design: 8/F, Knowles Building, The University of Hong Kong, Pokfulam Road, Hong Kong
*    Correspondence: waishanq@hku.hk or qiuwaishan@126.com; Tel.: +86-139-1655-6218

**Abstract:** Understanding park events and their categorization offers pivotal insights into urban parks and their integral roles in cities. The objective of this study is to explore the efficacy of Convolutional Neural Networks (CNNs) in categorizing park events through images. Utilizing image and event category data from the *New York City Parks Events Listing* database, we trained a CNN model with the aim of enhancing the efficiency of park event categorization. While this study focuses on New York City, the approach and findings have the potential to offer valuable insights for urban planners examining park event distributions in different cities. Different CNN models were tuned to complete this multi-label classification task, and their performances were compared. Preliminary results underscore the efficacy of deep learning in automating the event classification process, revealing the multifaceted activities within urban green spaces. The CNN showcased proficiency in discerning various event nuances, emphasizing the diverse recreational and cultural offerings of urban parks. Such categorization has potential applications in urban planning, aiding decision-making processes related to resource distribution, event coordination, and infrastructure enhancements tailored to specific park activities.

**Keywords:** urban park; human activity categorization; computer vision; publicly-available imagery

## 1. Introduction

Urban parks play a vital role in cities, and their importance to city residents has consistently grown over time. The benefits of urban parks include environmental benefits such as biodiversity and local cooling, economic benefits such as energy savings and property value, and social and psychological benefits such as physical activity and reduced obesity [1,2]. One of the important topics of park-related research is human events and programs in parks. Many studies have shown how park events could become a deciding force in shifting the park's own functionality [3–6]. In a report investigating London's urban parks, Smith and Vodicka [3] summarized from accounts of friends groups that events are seen as a promotion of the park's inclusivity that brings more people into the park, contributing to community cohesion. A similar study by Neal et al. [6] on parks also credits urban park events as an opportunity of inclusivity, as organized events present a more ethnically diverse population than regular park users. Citroni and Karrholm [7] analyzed the relationship of events to civility, drawing the conclusion that events facilitate the visibility of everyday life and forge a pattern of urban civility. Investigating events in urban parks is instrumental in understanding the multifaceted roles these green spaces play within city landscapes and their communities. Specifically, insights on park event distributions can illuminate the diverse contributions of urban parks, from cultural gatherings to educational activities. These insights can then further lead to the analysis of the influencing factors

and influences on park event distributions, which will aid urban planners in making informed decisions. This will ensure that park designs are not only reflective of the community's needs, but also actively enhance the quality of life, fostering health benefits and strengthening social ties.

There is a significant gap between existing works and efficient event analysis of the parks. Most of the past studies about park event analysis have focused on the intensity of park use [8–13], demographics of park users [14–16], the periods of time parks are used [14], and the level of physical activities [15,17]. However, few studies have been focusing on the categorization of park events and programs. From the aspect of data source, a majority of current studies analyzing the categories of human activities and planned events in parks have relied on mass questionnaires and interviews [18–21], which are time-consuming and site restrictive. Recent technological methods introduce big data into detailed park use analysis, such as GPS data and public participation geographic information systems data. However, GPS-based mobile phone tracking is not informative for the categorization of events and recreational park use [22], and public participation geographic information systems (PPGIS) cannot guarantee data sufficiency [23]. On the contrary, social media data and other publicly available online imagery are a good sources of information regarding recreational use of parks, and are thus a valuable resource for the purpose of this task. From the aspect of methodology, the methods of existing studies are either inefficient or not specifically targeted towards park events. Recent studies that utilize publicly available online imagery still involve tedious manual classifications [22]. The current research status calls for an updated methodology of a more accessible and cost-effective urban park events category analysis. This article utilizes the *New York City Parks Events Listing* [24] data, which is a set of publicly available, tagged image data, and proposes an algorithm featuring deep learning methods to more efficiently identify events and programming in urban parks. This is achieved through analyzing publicly available images of these parks, and performing classification based on park events. This is for the purpose of helping urban researchers and planners to better understand the impacts of park events in the community, and further incorporate them into the decision-making process.

*Related Works*

Although a significant number of studies have been conducted to determine the use of urban parks, the majority of these studies have focused quantitatively on the frequency or intensity of use [8–13]. Some emerging studies deploy crowd-sourcing visual survey to effectively collect public opinions (emotions and perceptions) on urban spaces including streetscapes and urban parks [25–32]. Some studies also investigated the demographics of park users [14–16], and the periods of time parks are used [14]. Regarding park activities, although a considerate number of studies have investigated the level of physical activity in parks [15,17], they identified simple events like sedentary, walking, or vigorous. Some studies went beyond this simple categorization and embodied a wider range of park activities [33,34]. However, more studies can still be done on a more fine-grained categorization of activities, as well as on activities driven by organized events as opposed to day-to-day activities such as walking or jogging. Lastly, it is also worth noting that many past studies on the use of urban parks focused on quantitatively examining the relationship between certain variables and the intensity of use. The independent variables examined include park proximity [8,9], park facilities [8], park quality [35], entrance fees [10], and social demographic characteristics of the neighborhood [8,10].

For the data source and methodology, traditional studies have relied heavily on questionnaires and personal interviews. For instance, Schipperijn et al. [18] conducted 14,566 face-to-face interviews with randomly-sampled Danish individuals, and asked these individuals to fill out follow-up questionnaires. Peschardt et al. [36] distributed 686 on-site questionnaires at nine small public urban green spaces to determine how these spaces were used by citizens. Nielsen and Hansen [9] mailed questionnaires to a sample of 2000 adult Danes. Other studies were conducted through direct observations in the parks.

For example, many studies, such as the ones by Marquet et al. [37] and Veitch et al. [38], employed the System for Observing Play and Recreation in Communities (SOPARC) [39] to directly observe residents' activities in parks. Similarly, Floyd et al. [15] measured physical activities in parks using a modified version of the System for Observing Play and Leisure Activity in Youth (SOPLAY). Brown et al. [33] used participatory GIS to investigate physical activities in urban parks. Overall, the application of traditional methods to understand park usages and park events is highly time-consuming and restrained to smaller areas due to the site-specificity [11].

Recent studies have been incorporating technologies to better understand the use of parks, both through utilizing novel online data sources and more efficient categorization. Commonly used novel data sources include social media data, geo-tracking data from mobile phones, and PPGIS data. For instance, Li et al. [12] retrieved geo-tagged social media check-in records for park visits to examine the frequency of visits. A bivariate correlation analysis was conducted to support the association between the Weibo check-in data and official visitor statistics, although the strength of correlation ranges from city to city. Larson et al. [11] used geo-tracking data from cell phones to document changes in park visits during the COVID-19 pandemic. Heikinheimo et al. [22] compared four types of data (social media, sports tracking, mobile phone operator and PPGIS data) in a case study of Helsinki, Finland, and examined the ability of these user-generated datasets to provide information on the use of urban parks.

In comparison, social media data is highly informative for the leisure time activities being conducted in urban parks [22], but is limited by biases in age groups and the choice to share content publicly [40]; mobile phone data highlights movements [22], but only best represents populations in countries where mobile phones are widely used [41]; PPGIS allows the researcher to ask in-depth questions on park use and preferences [22], but the response rate and its fairness are not guaranteed [23].

For categorization methods, the content analysis of social media data in Heikinheimo's study was done through manual classification of 15,312 Instagram photos and 1843 Flickr photos. This is again time-consuming and inefficient, and calls for a more automatic method of analyzing social media content on park activities. To compare the best-known commercial image recognition service providers on this task, Ghermandi et al. [34] performed a test using Google Cloud Vision [42], Clarifai [43], and Microsoft Azure Computer Vision [44] to identify human-nature interactions (outdoor recreational activities, biophysical environments, and feelings) in parks. All of these models surpass traditional methods in the efficiency of categorization. However, due to the generic nature of the image recognition services, the tags identified in relation to recreational activities are relatively limited, without sufficient specificity to park-related, event-driven activities. For example, all three services identified people posing for a photograph as the most frequent activity captured in social media imagery. Another precedent to this study is Matasov et al.'s study on COVID-19's impact on the recreational use of Moscow parks, which applied the YOLOv5x neural network to conduct object detection on geo-tagged social media photos [45].

In conclusion, there are three research gaps in the existing research. Firstly, current studies focus more on the intensity of park usage and level of physical activities (sedentary, walking, vigorous), leaving a gap for more fine-grained studies in the categorization of park events. Secondly, for the methodology, traditional studies rely heavily on questionnaires and personal interviews, which are time-consuming and restricted. Thirdly, in recent studies that incorporate technologies, the categorization methods are either inefficient or not specific to park events. To fill the current research gaps, this study contributes to the literature in these following ways: by focusing the analysis on the categorization of park events; by incorporating the use of publicly available imagery to increase the efficiency of analysis; and by proposing transfer learning on pre-trained Convolutional Neural Networks (CNNs) to calibrate the model towards the park event identification task, achieving a 0.876 accuracy and a 0.620 mean average precision. Table 1 summarizes the current research gaps and this study's relative contributions.

**Table 1.** Research Gaps and Contributions.

| Research Gap | Contribution |
|---|---|
| Current studies focus on intensity of park use | Our study focuses on categorization |
| Current studies rely on questionnaires and personal interviews | Our study incorporates use of publicly available imagery to increase efficiency |
| Technological methods in current studies are either inefficient or not specific to park events | Our study proposes transfer learning on pretrained CNNs |

## 2. Dataset and Methods

### 2.1. Research Framework

To more efficiently identify events in urban parks, this research applies Convolutional Neural Networks (CNNs) on images in the *New York City Parks Events Listing* [24] database to conduct multi-label classification of park events. Firstly, we conduct data preprocessing, as described in Section 2.3. Preprocessing includes two aspects: clustering the event categories and applying transfer learning to remove all non-photographic visual media. The images utilized from the *New York City Parks* website were also resized to a resolution of 224 × 224 pixels and adjusted to RGB mode for the purposes of our study. Secondly, we compared across different machine learning models to determine the best model for the multi-label classification task. Models examined include VGG16 [46], ResNet50 [47], ResNet18 [47] and GoogLeNet [48]. See Figure 1 for the overall research methodology.

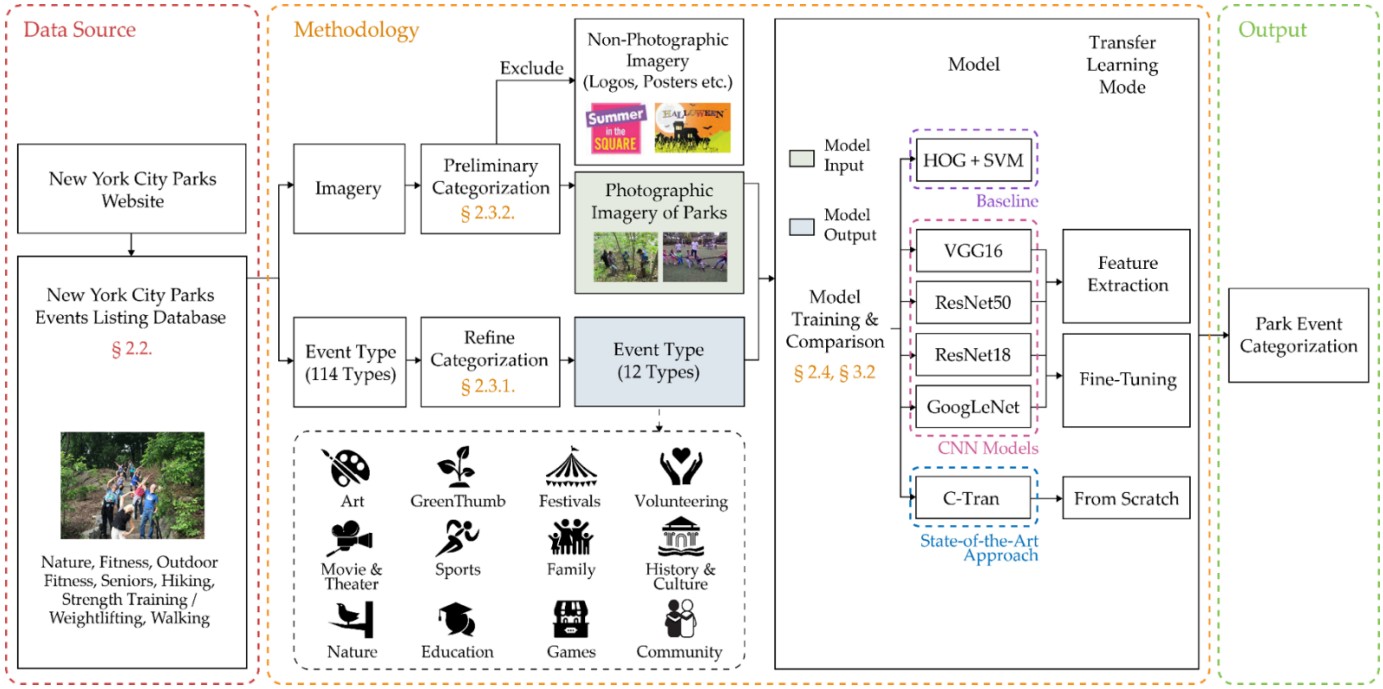

**Figure 1.** Overall Research Methodology.

### 2.2. Dataset

The models are trained on the *New York City Parks Events Listing* database. This database is used to store event information displayed on the *New York City Parks* website, https://www.nycgovparks.org/ (accessed on 18 August 2023) [49], which displays events from parks all over New York City. See Figure 2. This includes "more than 5000 individual properties ranging from Coney Island Beach and Central Park to community gardens and Greenstreets" [49]. The *New York City Parks Events Listing* database contains the title, date, time, location, description, contact information, categories, and images of the events since

2013. In total, it contains 11,060 event images, which are linked to 114 event categories. This contains event records from 2013 through 2 August 2021.

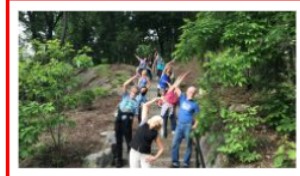

**Figure 2.** A capture of the *New York City Parks* website, with red boxes highlighting the two crucial data points used in our study: the event image and its associated categories.

*2.3. Data Preprocessing*

For the purpose of this study, we were only extracting the images and event category information from the dataset, using the Event IDs to link the two together. There were two issues with the original dataset: different levels of specificity in the event categories and the inclusion of non-photographic imagery (logos, posters etc.). Preprocessing was performed to further refine the categorization, reduce the noise, and increase generalizability.

2.3.1. Refining the Categorization

The first issue with the dataset was that the 114 different categories of events in the dataset had different levels of specificity. Some categories were very general, such as "Nature", "Art" or "Volunteer". Other categories were as specific as "Brooklyn Beach Sports Festival" or "MillionTreesNYC: Volunteer: Tree Stewardship and Care". This presented a challenge in ensuring consistent granularity when analyzing or interpreting the data. Therefore, during preprocessing, we manually grouped these categories into larger groups and formed twelve new categories. See Table 2. Our final categories were determined by reviewing previous studies on park events and their employed categorizations. Eleven out of our twelve categories have been explored, to varying degrees, in prior research: Art [3], GreenThumb [3], Festivals [3], Volunteering [3], Film [3], Sports [3], Family [50], History & Culture [3], Nature [3], Education [51], and Community [3]. The category 'Games' served as a catch-all for events related to games that don't easily fit within the broader categories. It is worth noting that during our regrouping process, we refrained from mapping a single original category to multiple final categories. This decision was informed by the inherent characteristics of the dataset, where an individual event may be classified under multiple categories, leading to a degree of redundancy in labeling. For instance, while "Historic House Trust Festival" might naturally fit both "Festivals" and "History & Culture", we observed that every event tagged as "Historic House Trust Festival" also carried the "Historic House Trust Sites" label. Since "Historic House Trust Sites" was re-categorized to "History & Culture", we chose to exclusively map "Historic House Trust Festival" to "Festivals".

**Table 2.** Event Categorization.

| Final Category | Original Category |
|---|---|
| Art | Art, Arts & Crafts, Art in the Parks: Celebrating 50 Years, Art in the Parks: UNIQLO Park Expressions Grant |
| GreenThumb | GreenThumb Events, GreenThumb Partner Events, GreenThumb 40th Anniversary, GreenThumb Workshops |
| Festivals | Festivals, Historic House Trust Festival, Valentine's Day, Halloween, Saint Patrick's Day, Earth Day & Arbor Day, Mother's Day, Father's Day, Holiday Lightings, Santa's Coming to Town, Lunar New Year, Pumpkin Fest, Summer Solstice Celebrations, Easter, Fall Festivals, New Year's Eve, Winter Holidays, Thanksgiving, National Night Out, Black History Month, Women's History Month, LGBTQ Pride Month, Hispanic Heritage Month, Native American Heritage Month, Fourth of July, City of Water Day, She's On Point |
| Volunteering | Volunteer, MillionTreesNYC: Volunteer: Tree Stewardship and Care, Martin Luther King Jr. Day of Service, MillionTreesNYC: Volunteer: Tree Planting |
| Film | Film, Free Summer Movies, Theater, Free Summer Theater, Movies Under the Stars, Concerts, Free Summer Concerts, SummerStage, CityParks PuppetMobile |
| Sports | Fitness, Outdoor Fitness, Running, Bike Month NYC, Hiking, Learn To Ride, Sports, Kayaking and Canoeing, National Trails Day, Brooklyn Beach Sports Festival, Summer Sports Experience, Fishing, Girls and Women in Sports, Bocce Tournament |
| Family | Best for Kids, Kids Week, CityParks Kids Arts, School Break, Family Camping, Dogs, Dogs in Parks: Town Hall, Seniors, Accessible |
| History & Culture | History, Historic House Trust Sites, Arts, Culture & Fun Series, Shakespeare in the Parks |
| Nature | Nature, Birding, Wildlife, Wildflower Week, Cherry Blossom Festivals, Waterfront, Rockaway Beach, Bronx River Greenway, Fall Foliage, Summer on the Hudson, Living With Deer in New York City, Tours, Freshkills Tours, Freshkills Park, Urban Park Rangers, Reforestation Stewardship |
| Education | Talks, Education, Astronomy, Partnerships for Parks Tree Workshops |
| Games | Dance, Games, Recreation Center Open House, NYC Parks Senior Games, Mobile Recreation Van Event |
| Community | Open House New York, Community Input Meetings, Fort Tryon Park Trust, Poe Park Visitor Center, Shape Up New York, City Parks Foundation, Forest Park Trust, City Parks Foundation Adults, Partnerships for Parks Training and Grant Deadlines, Community Parks Initiative, Anchor Parks, Markets, Food |

### 2.3.2. Remove Non-Photographic Imagery

The second issue with the dataset was that it was a mix of photos taken at the parks, and non-photographic visual media such as posters of events and logos of host organizations. To resolve this issue, we introduced feature extraction transfer learning during preprocessing to conduct binary classification and remove the non-photographic images. We applied a VGG16 [46] model pre-trained on the ImageNet [52] dataset, freezing its base layer weights and adding a custom sigmoid layer on top to conduct binary classification. After the top layer was trained on 640 manually-labeled images from the dataset for 25 epochs, with an Adam optimizer and a learning rate of 0.0003, the model achieved a 0.88 training accuracy and a 0.92 accuracy on 160 labeled test images. With this highly accurate model, we can apply it on the entire dataset to filter out non-photographic images as predicted. This reduced the dataset size from 11,060 images to 7427 photos.

### *2.4. Classification Modeling*

### 2.4.1. Model Selection

A wide range of machine learning models was examined in this study to determine the best model for this task, where the inputs were event images and the expected outputs were predictions of the categories of the event.

1. Baseline: Histogram of Oriented Gradients (HOG)—Support Vector Machine (SVM) based model

A Histogram of Oriented Gradients (HOG) feature is a feature descriptor used in computer vision and image processing for object detection [53]. The Support Vector Machine (SVM) is a supervised learning algorithm commonly used for classification tasks [54]. A combination of HOG and SVM is incorporated in this study as an example of a traditional approach, where HOG features are extracted from the images and classified through the SVM.

2. Convolutional Neural Networks (CNNs) based models

Convolutional Neural Network (CNN) is a class of artificial neural networks most commonly applied to analyze visual imagery [55]. This study incorporated a selected range of classic CNN models such as VGG16 [46], ResNet50 [47], ResNet18 [47] and GoogLeNet [48]. For each of these CNN models, custom layers including an average pooling layer, a dense layer of 32 neurons (ReLU activation), and a dense layer of 12 neurons (sigmoid activation) were incorporated on top to conduct multi-label classification. The sigmoid layer replaces the conventional softmax layer to accommodate the presence of multiple labels per input image (a park could be used for both fitness and birdwatching). Softmax gives a probability distribution over the entire span of classes, where the 12 probabilities for 12 classes add up to one. By using sigmoid instead, we give each class a number between 1 and 0, and the probabilities do not have to add up to one. Thus, the probability of picking one class is independent of other classes, and we may have multiple labels.

3. State-of-the-Art Approach: C-Tran

C-Tran [56] is a recently proposed model by Lanchantin et al. in 2021, which utilizes Transformers for multi-label image classification. In this study, C-Tran is included as an exemplar of the latest approaches in solving the multi-label classification problem. However, there are limitations to the application of C-Tran in our study due to the discrepancy between the full-image categorization nature of our dataset and the specific dataset assumptions of C-Tran. This is further detailed in Section 2.4.2.

### 2.4.2. Training

The training process was conducted in the Google Colab environment, using TensorFlow 2.12.0 and a V100 GPU. The models were trained on 80% of the images, with the remaining 20% retained for validation and model assessment. Hyperparameters for all

CNN models were generally determined through tuning on the VGG16 model, which generated a group of optimized values (batch size = 64, learning rate = 0.0002, number of epochs = 80). These hyperparameters for certain models were slightly tuned in later training. See Table 3. For example, ResNet18 with a batch size of 64 generated suboptimal results. A test of 10 epochs was conducted among ResNet18 models being fine-tuned with batch sizes of relatively 64, 32 and 16, which determined that 32 was the most optimized. All CNN models and C-Tran used the Adam optimizer.

In this study, transfer learning was particularly chosen due to its advantages in efficiency and performance. Training deep neural networks from scratch would require significant computational resources and might not leverage the rich feature-learning already established in networks trained on datasets like ImageNet. Given the specific context of our park events dataset, which is much smaller and more specialized than vast datasets like ImageNet, it was essential to capitalize on the foundational features such networks have already discerned, like textures or shapes that might be common in park images. Initializing our models with weights from a network pre-trained on ImageNet not only accelerates the training process but also helps in achieving better convergence. Additionally, using transfer learning mitigates the risk of overfitting, especially crucial when working with limited datasets. Accordingly, for each of the CNN models, both feature extraction and fine-tuning techniques were employed for testing. Feature extraction involves freezing the pretrained base layer weights during training, while in fine-tuning all layers are made trainable. The performances of these techniques were then compared to discern the optimal approach for our dataset.

**Table 3.** Hyperparameters for model training.

| Model | Transfer Learning Mode | Batch Size | Learning Rate | Epochs |
|---|---|---|---|---|
| VGG16 | Feature Extraction | 64 | 0.0002 | 80 |
| | Fine-Tuning | 64 | 0.0002 | 80 |
| ResNet50 | Feature Extraction | 64 | 0.0002 | 100 |
| | Fine-Tuning | 64 | 0.0002 | 70 |
| ResNet18 | Feature Extraction | 32 | 0.0002 | 20 |
| | Fine-Tuning | 32 | 0.0001 | 10 |
| GoogLeNet | Feature Extraction | 64 | 0.0002 | 80 |
| | Fine-Tuning | 64 | 0.0002 | 60 |
| C-Tran | From Scratch | 1 | 0.00001 | 40 |

For the C-Tran model, the event description feature from the event listing database was extracted as the image caption for the event image, which should be noted as a limited approach, as the algorithm was originally designed assuming the caption to be a clear and concise description of the image content.

### 2.4.3. Evaluation Metrics

This study incorporates both the accuracy and the mean Average Precision (mAP) metrics to evaluate the model performance. In calculation of the accuracy, we treat the classification of each model as an independent task, and calculate the average accuracy across labels. We also incorporated the mAP, a commonly used metric to evaluate object detection models, as it is a relatively comprehensive evaluation metric that takes into account both precision and recall for each class or label.

## 3. Results

### 3.1. Descriptive Statistics

Figure 3 shows the distribution of images across different labels in the dataset after non-photographic imagery was removed (as described in Section 2.3.2). 'Family', 'Nature', and 'Film' are the three categories that occurred most frequently. 'GreenThumb' and 'Volunteer' only contain a very small number of images.

Figure 4 presents all parks and green spaces in New York City being analyzed. Figure 5 illustrates the distribution of event categories within New York City parks. For a visual representation of this distribution in Figure 5, we aggregated all events situated within the boundaries of each park. Then, circles are positioned at the centroid of each park's geometry or, if a park has multiple geometries, the centroid of the aggregated geometries associated with that park. The number of events for each category are indicated by both the color and the size of these circles.

Events categorized under 'Film' are prevalent across numerous locations, suggesting that many of these parks are equipped for outdoor film screenings or theatrical performances. The event categories of 'Family', 'Festivals', 'Games', and 'Community' also demonstrate a relatively uniform distribution across parks. Whether it's many parks hosting a high volume of such events or a minimal variance between parks with the most and least of these events, such distribution suggests that these activities generally necessitate less specialized infrastructure or equipment. Conversely, while the 'Art' category displays a peak value of 249 events at a single park, such events are less widespread. This limited distribution indicates that specialized facilities are needed for art events, possibly making them less accessible to residents citywide. Similarly, parks housing 'History & Culture' events, apart from the notable Central Park, predominantly include history-centric venues such as Roger Morris Park and Alice Austen Park. Parks featuring 'Nature' events are predominantly located towards the city's outskirts, a placement that seems intuitive given the larger, more natural landscapes in those regions. 'GreenThumb' and 'Volunteering' events exhibit a more selective distribution, with a handful of parks like Windmill Community Garden for the former, and iconic spots like Central Park and Prospect Park for the latter, emerging as predominant hosts. 'Education' events are also highly concentrated in specific parks, namely Central Park, Wave Hill Public Garden and Cultural Center, as well as Conference House Park. 'Sports', on the other hand, presents an intriguing pattern; while certain parks are hotspots, the distribution seems less governed by the presence of sports facilities and more influenced by factors like local community culture, park location, and proximity to organizations that might host fitness classes and related activities.

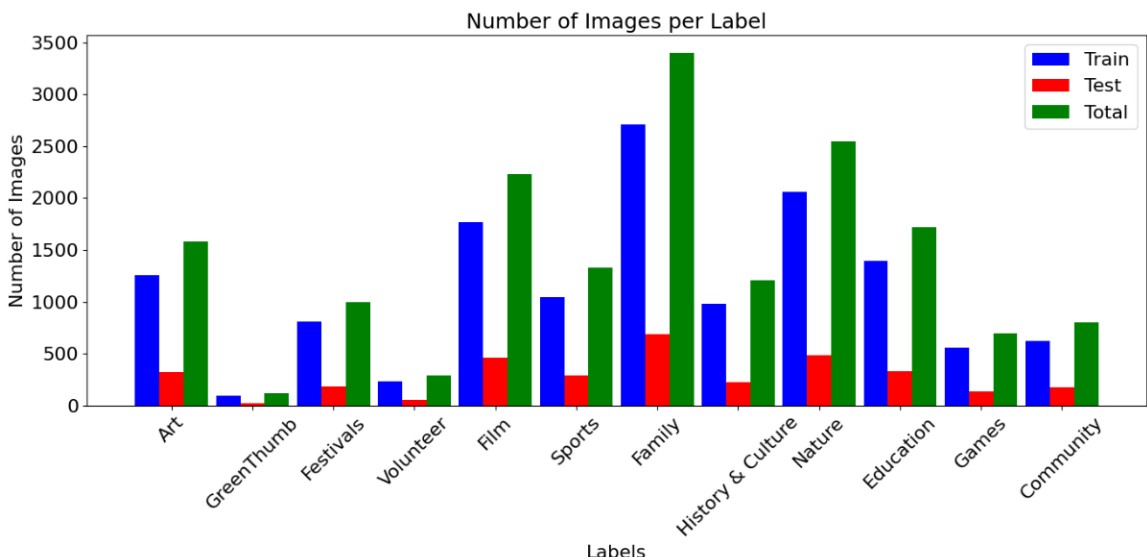

**Figure 3.** Image distribution across event types.

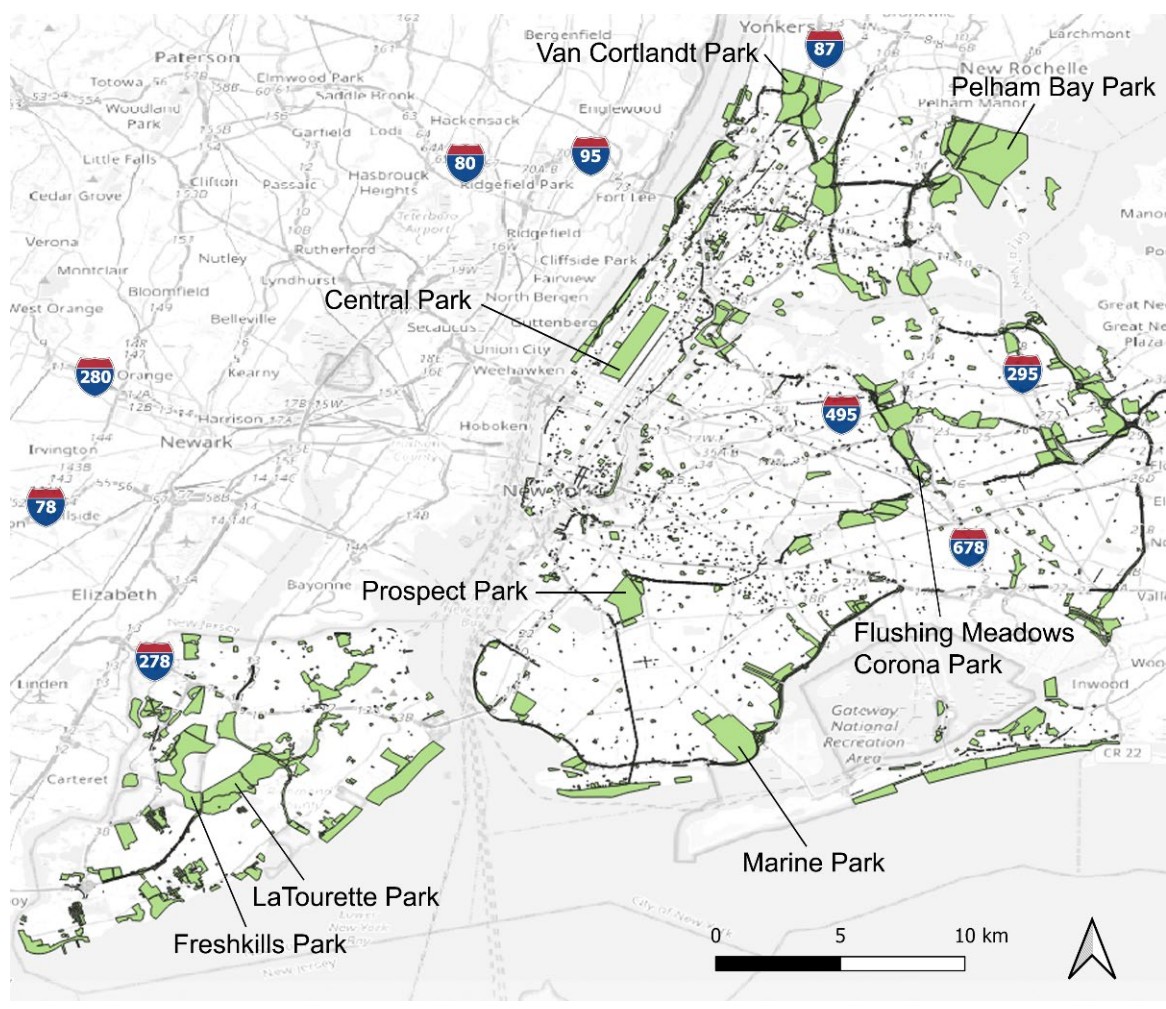

**Figure 4.** New York City green spaces. Selected flagship parks are labeled with names.

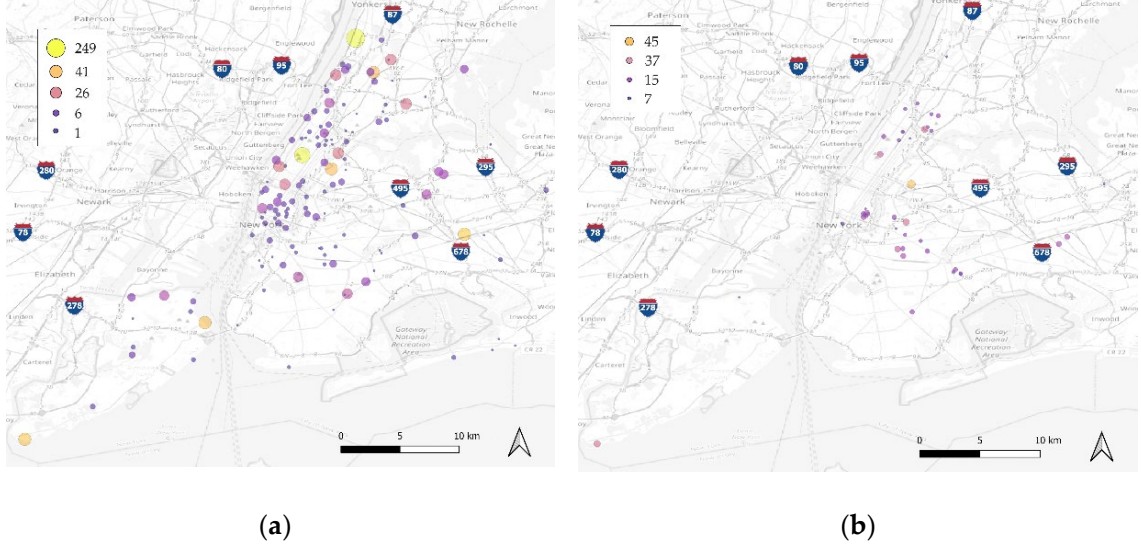

(**a**)          (**b**)

**Figure 5.** *Cont.*

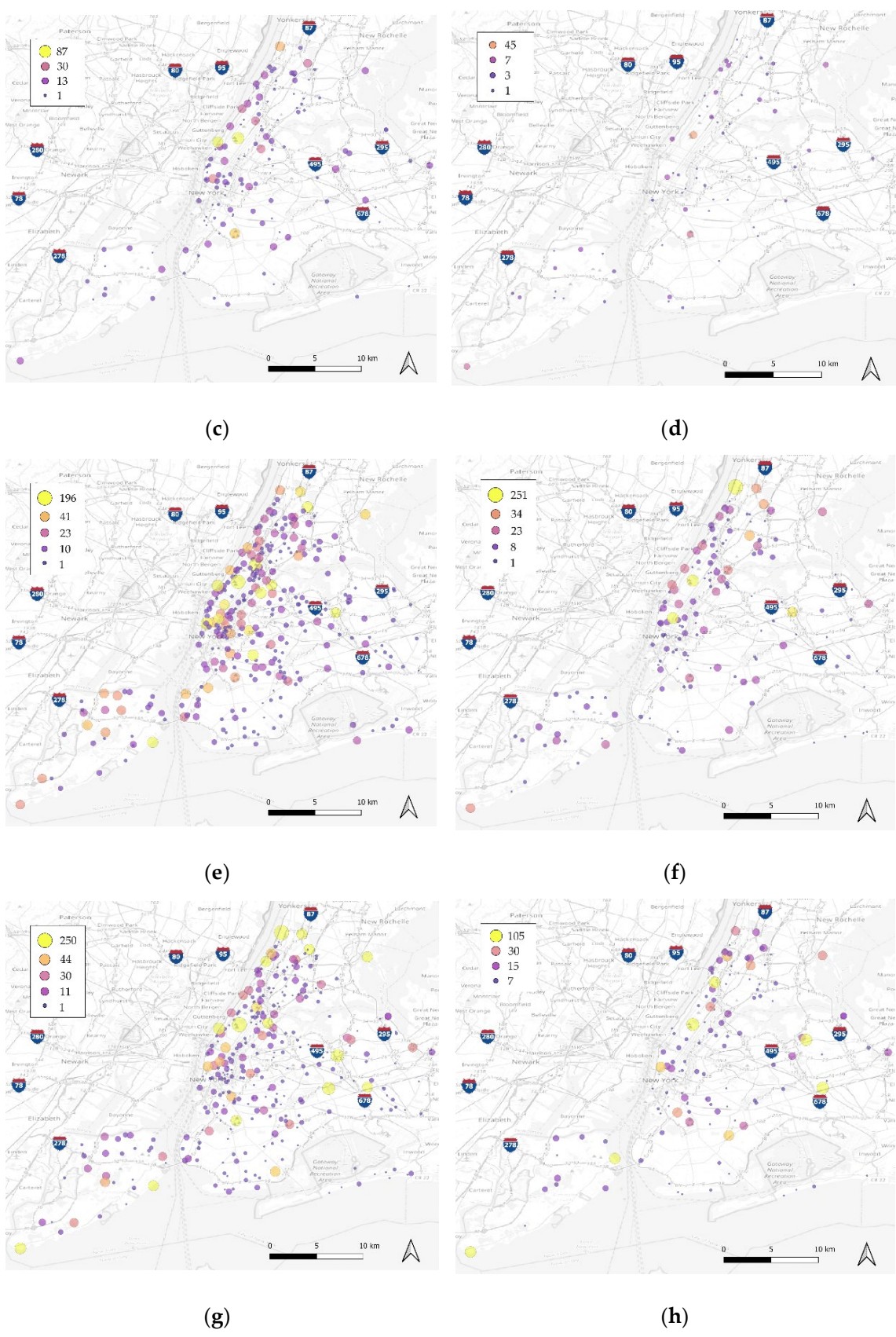

(**c**)

(**d**)

(**e**)

(**f**)

(**g**)

(**h**)

**Figure 5.** *Cont.*

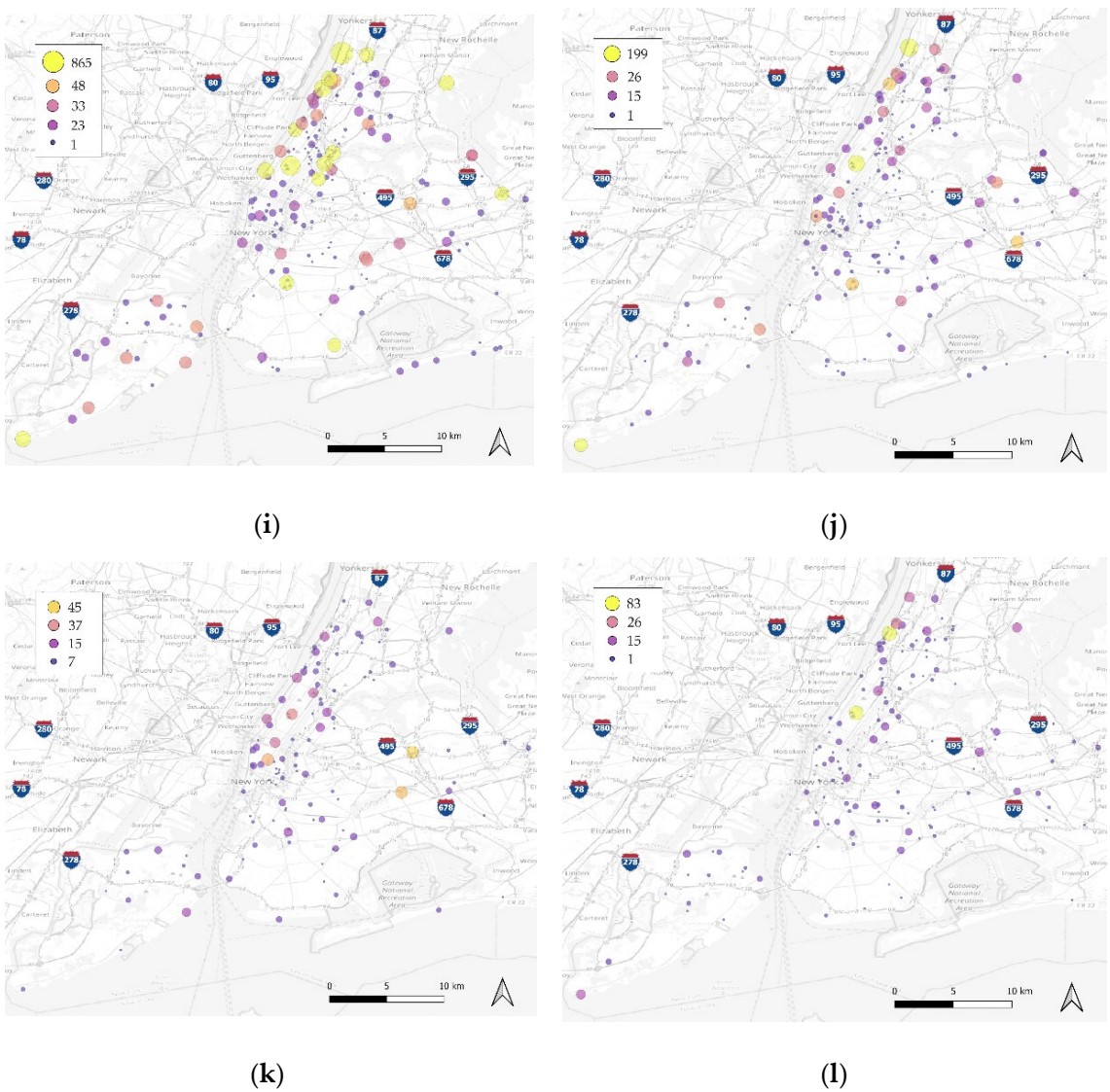

**Figure 5.** Distribution of park event categories across parks in New York City. (**a**) Art. (**b**) Green-Thumb. (**c**) Festivals. (**d**) Volunteering. (**e**) Film. (**f**) Sports. (**g**) Family. (**h**) History & Culture. (**i**) Nature. (**j**) Education. (**k**) Games. (**l**) Community.

In the diverse urban tapestry of New York City, parks emerge as dynamic spaces of community interaction and learning. Figure 6 shows the co-occurrence matrix of different event types. We observed that 'Family' and 'Art', 'Family' and 'Film', 'Family' and 'Nature', 'Family' and 'Education', and 'Nature' and 'Education' are frequent co-occurrences. The co-occurrence of events such as 'Family & Art' underscores the city's commitment to fostering a vibrant arts culture, making it accessible to audiences of all ages. Outdoor movie sessions, exemplified by the 'Family & Film' pairing, showcase the parks' ability to transform into open-air theaters, creating unique urban experiences. The conjunction of 'Family & Nature' and 'Family & Education' emphasizes the parks' role as both recreational escapes and vital educational hubs. Parks not only offer families a chance to reconnect with nature but also provide hands-on educational experiences. Lastly, the overlap between 'Nature & Education' reiterates the importance of these urban green spaces in fostering environmental awareness and stewardship among its citizens. Such multifaceted interactions in New York City parks highlight their indispensable role in enhancing the city's cultural, recreational, and educational landscape.

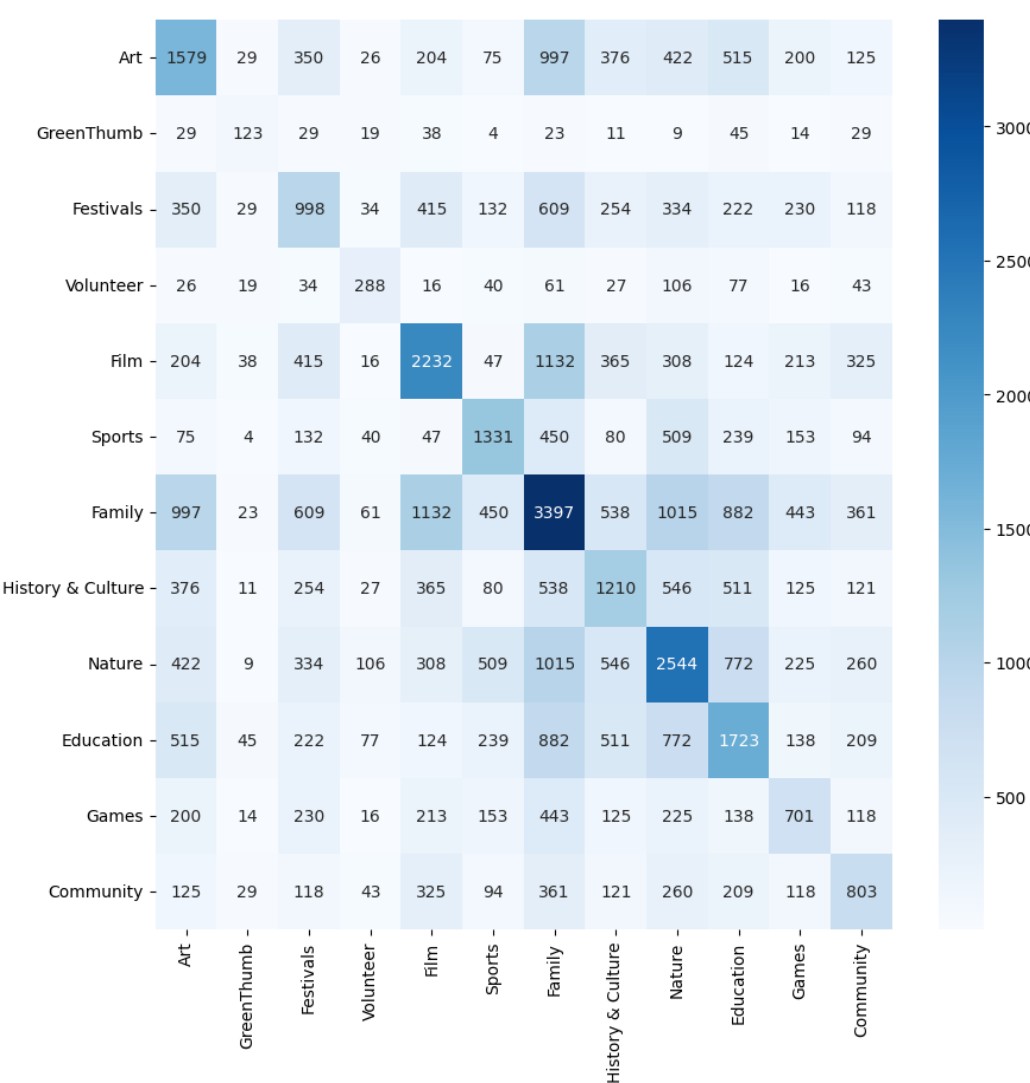

**Figure 6.** Event type co-occurrence matrix.

### *3.2. Overall Performance of Event Classification*

Figure 7 presents the accuracy and mean Average Precision change throughout the training process for both feature extraction and fine-tuning on ResNet50, as an example comparison for these two transfer learning approaches.

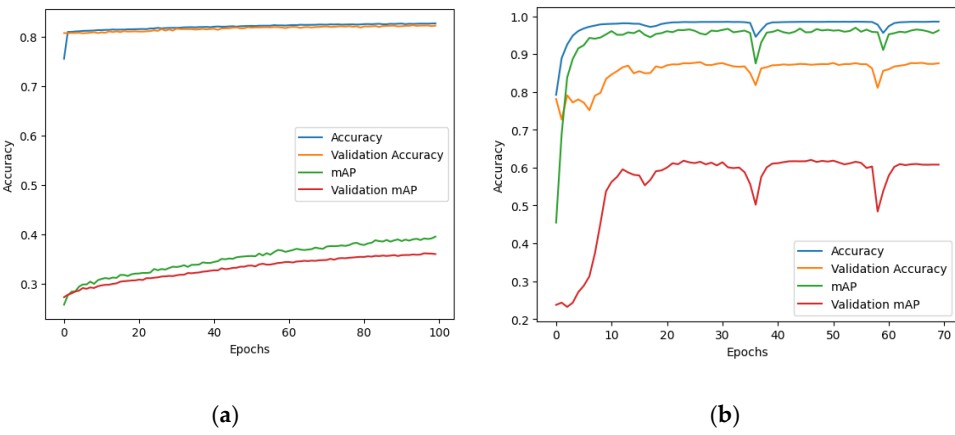

(**a**)                                             (**b**)

**Figure 7.** Comparing accuracy and mean Average Precision between different transfer learning approaches. (**a**) ResNet50 Feature Extraction. (**b**) ResNet50 Fine-Tuning.

Table 4 presents all results from the models examined, including the baseline HOG + SVM approach and the state-of-the-art C-Tran model. Among all the examined approaches, fine-tuning on the ResNet50 model achieved the best performance in both accuracy and mean Average Precision, outperforming ResNet18 and GoogLeNet (InceptionV3) by a small margin. This suggests that ResNet50 was the most capable in capturing the features that indicate park events and recreational human activities in this dataset.

**Table 4.** Validation Accuracy and mAP.

| Model | Transfer Learning Mode | Accuracy | mAP * |
|---|---|---|---|
| HOG + SVM | From Scratch | 0.861 | 0.345 |
| VGG16 | Feature Extraction | 0.844 | 0.462 |
| | Fine-Tuning | 0.854 | 0.564 |
| ResNet50 | Feature Extraction | 0.823 | 0.360 |
| | Fine-Tuning | **0.876** | **0.620** |
| ResNet18 | Feature Extraction | 0.809 | 0.291 |
| | Fine-Tuning | 0.870 | 0.601 |
| GoogLeNet | Feature Extraction | 0.857 | 0.551 |
| | Fine-Tuning | 0.876 | 0.602 |
| CTran | From Scratch | - | 0.200 |

* mean Average Precision.

Figure 8 presents the normalized confusion matrices for each label, where the $x$ axis is the prediction (with a threshold of 0.5) and the $y$ axis is the ground truth. These graphs show that for all labels, true negatives compose the majority of the confusion matrices, and false positives compose the least percentage. This suggests that the model is generally conservative in its predictions. There are missed opportunities in the labels 'GreenThumb', 'Festivals', 'Volunteer', 'History & Culture', 'Education', 'Games', and 'Community', where false negatives outnumber true positives. Among these labels, 'GreenThumb' (99) and 'Volunteer' (233) are labels with a very low portion of corresponding training images. 'Festivals' (809), 'History & Culture' (984), and 'Education' (1393) are labels with relatively sufficient training images, but still exhibit a concerning number of false negatives, which suggests that the model's inability to accurately predict these categories is potentially due to other factors such as data quality and label ambiguity. 'Games' (560) and 'Community' (625) are labels with a medium number of images, and the cause of underperformance is hard to determine. The model is particularly successful in predicting the presence of 'Film', 'Family', and 'Nature'. These are also the three categories that compose the overwhelming majority of the training dataset, with each category containing more than 1700 images.

Another contributing factor for the accurate identification of events under the 'Film', 'Family', and 'Nature' categories could be the distinct features found within the parks themselves. These unique amenities or landmarks may be intrinsically tied to the events in these categories. For instance, parks hosting 'Film' events may have dedicated open spaces or amphitheaters suitable for large audiences, those emphasizing 'Family' events might possess playgrounds or picnic areas designed for family gatherings, and parks with frequent 'Nature' events could be characterized by trails, water bodies, or other natural landmarks. Such distinct features could make categorizing events in these parks more straightforward.

Figure 9 presents the normalized co-occurrence matrix for the true and predicted labels, where the $x$ axis represents the predicted classes, and the $y$ axis represents the true classes. On the diagonal, 'Film', 'Sports', 'Nature' and 'Family' are the four labels with the highest percentage of successful classification. 'Festivals' is a label that the model

specifically struggles with. It is also worth noting that, due to the multi-label nature of the classification task, the ideal for this matrix is not necessarily to have high values only along the diagonal. For example, high values occur on the intersections of the 'Family' row and the 'Art', 'Film', 'Nature' and 'Education' columns. This is exactly in correspondence with what we observed in Section 3.1. about the co-occurrences in the dataset, potentially suggesting that the model was successful in identifying genuine patterns from the data.

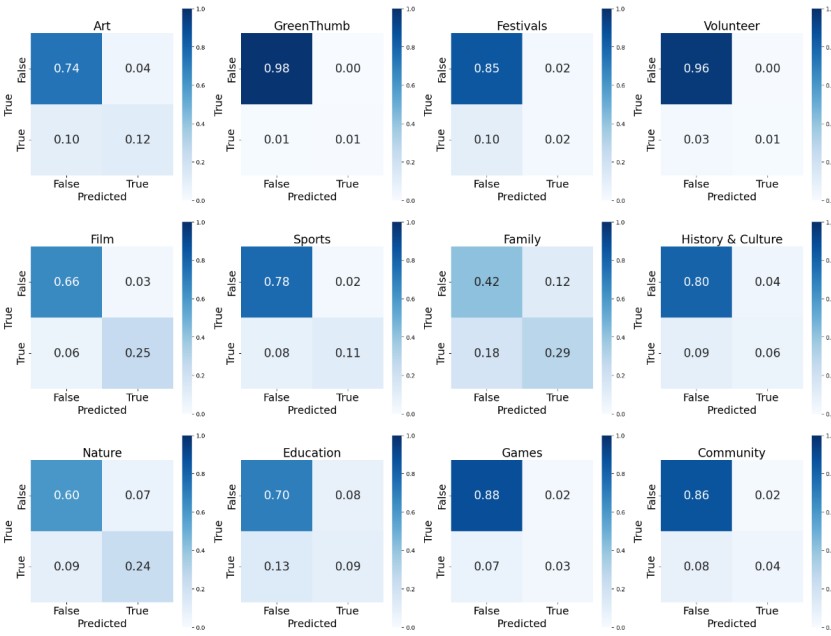

**Figure 8.** Normalized confusion matrices (*X* axis = predicted classes; *Y* axis = true classes).

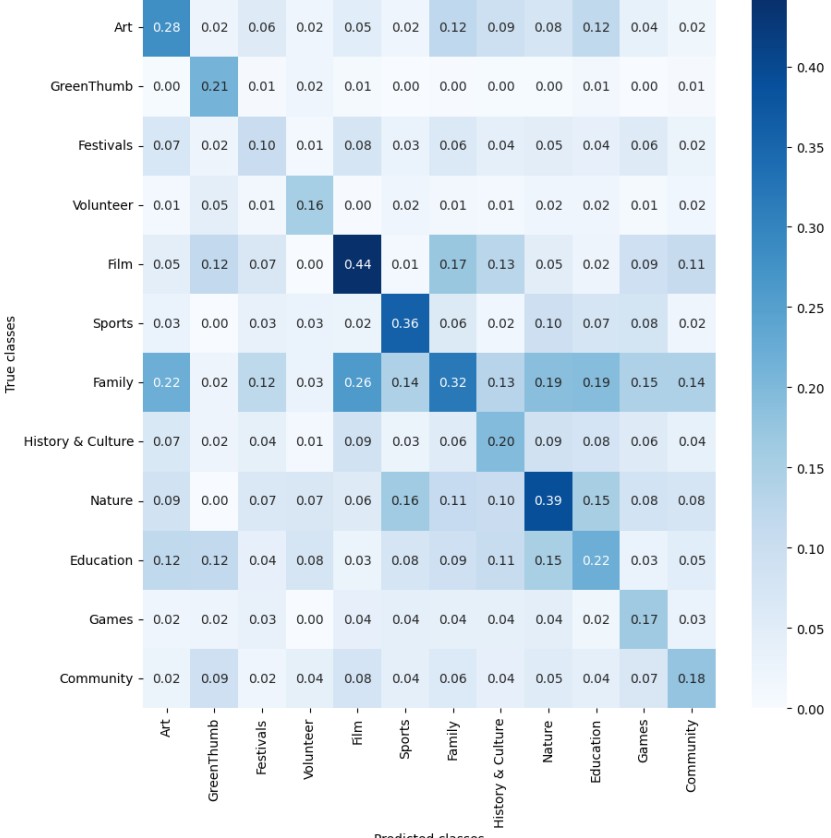

**Figure 9.** Event type co-occurrence matrix (*X* axis = predicted classes; *Y* axis = true classes).

### 3.3. Transfer Learning Approaches

It is worth noting that for this task, fine-tuning on all CNN models outperforms feature extraction transfer learning, and some models such as ResNet18 even showed significant performance differences. This might suggest that there is a limited similarity between the task of the pre-trained model (object recognition based on ImageNet) and the target domains of this task. This can be attributed to the nature of the dataset where in a lot of the images, the model needs to recognize the gesture of the human(s) to determine the label, while the ImageNet dataset is organized only around nouns [52]. Another thing this suggests is the complexity of the task, since the situation could indicate that the task's complexity exceeds what can be adequately addressed by the feature extraction approach. Fine-tuning, on the other hand, allows the model to adapt to the specific features of the New York City park event images. Figure 10 presents examples of park event images, their true labels and predicted labels. This offers a tangible representation of the model's predictive capabilities, showcasing instances where the model successfully identified the event type as well as moments of misclassification. By observing the images side-by-side with their labels, readers can gain insights into the nuanced features the model potentially considers when making its predictions.

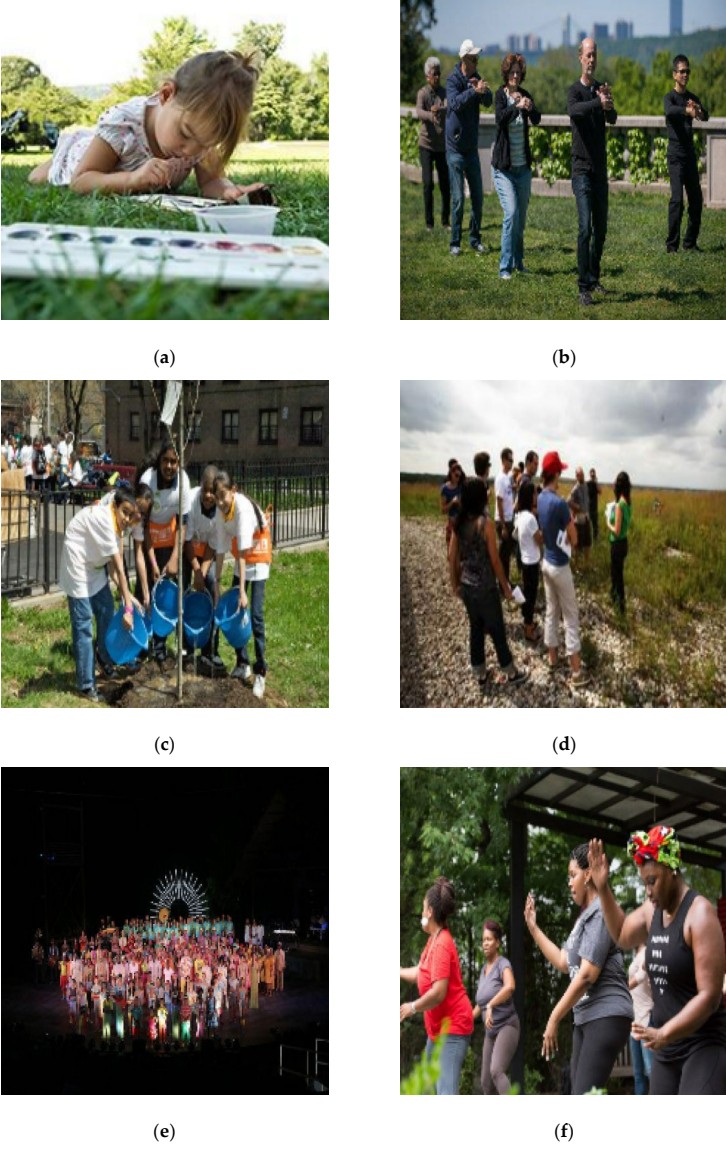

(a)  (b)

(c)  (d)

(e)  (f)

**Figure 10.** Example images, their true labels and predictions from ResNet50 Fine-Tuning. (**a**) True: {*Art*, *Family*}; Predicted: {*Art*, *Family*, *Education*}. (**b**) True: {*Sports*}; Predicted: {*Sports*}. (**c**) True: {*GreenThumb*,

*Volunteer, Education*}; Predicted: {*GreenThumb, Volunteer, Education*}. (**d**) True: {*Nature, History & Culture*}; Predicted: {*Nature, Family, Volunteer*}. (**e**) True: {*Art, Festivals, Film, History & Culture*}; Predicted: {*Art, Festivals, Film, History & Culture*}. (**f**) True: {*Art, Sports, Games*}; Predicted: {*Sports, Games*}.

## 4. Conclusions

Understanding park events and being able to categorize them is crucial to understanding parks and their role in urban areas. This study uses the images and event category information in the *New York City Parks Events Listing* database to train a Convolutional Neural Network that categorizes park events represented in images. Upon evaluating various models, it was determined that ResNet50 emerged as the most proficient in the event categorization task, achieving an accuracy of 0.876 and a mAP of 0.620, outperforming the other models compared. The results demonstrate the potential of deep learning techniques in automating the categorization process of park events, which can provide invaluable insights into the activities and cultural dynamics within urban parks.

This study holds notable significance, particularly in enhancing research and planning related to urban park use. While scholars have demonstrated a sustained interest in exploring urban park use—investigating aspects from health effects [9,10] to determinants of park use [8,14,18–20]—their methods have often been constrained by the inefficiencies of direct observation [15,17,37,39], interviews [18], and mass questionnaires [9,18,36]. The Convolutional Neural Network (CNN) method proposed herein provides a promising alternative, capably recognizing and distinguishing a wide array of activities within urban parks, thereby equipping researchers with a potent tool for future investigations. Furthermore, accurate categorization can aid city planners and park administrators in making informed decisions about resource allocation, event scheduling, and infrastructure development tailored to the unique needs of different event types. This is applicable in both the design and operational phases of park management. For instance, accurate event categorization can unveil the mismatches between existing park facilities and the prevalent types of events. To illustrate, the Union Square Park is a relatively compact space nestled between Park Avenue and 5th Avenue in New York City. Despite the absence of any sports-related facilities, 'Sports' emerged as the most predominant event type in the park, with 72 events labeled sports-related, many of which are fitness classes. This is notably higher than in other parks on the periphery of New York City, which are equipped with diverse sports facilities like tennis and basketball courts. Factors such as location convenience and community preference might have influenced this trend. Consequently, planners may contemplate integrating more facilities to accommodate fitness activities in Union Square Park. Additionally, such insights can shape the design of future parks, ensuring they are aptly equipped to support desired activities.

Future avenues of research encompass both the application of our trained model to unlabeled datasets and the expansion of our labeled datasets to further hone the model's accuracy. To begin with, our model can be deployed on unlabeled datasets from popular social media platforms like Instagram and Flickr. This would enable efficient categorization of park-related event images, providing deeper insights into event distributions. To further enhance our understanding of the diverse roles urban parks play within communities and inform urban planning, several other analyses can be performed taking advantage of the park event distribution information. Two main directions of further study include the influencing factors of park event distribution and the influence of that distribution. For the former, future studies can inspect a variety of factors influencing the range of events happening in parks, including size of the park, culture and demographics of surrounding areas, park facilities, vegetation, open space, and orography. This will help planners better design the physical space in order to encourage certain types of events. For the latter, with the help of this study, future studies can quantitatively examine the relationship between park events and their social benefits such as health benefits and increased social

cohesion. This will inform planners on what activities will be most beneficial for certain areas, and design parks to encourage these activities. Lastly, integrating more labeled data, sourced from similar park event listing websites such as the one from Millennium Park in Chicago [57], can bolster the model's performance, ensuring more accurate and robust categorizations in future applications. To maintain consistent model performance on the expanded dataset, additional experiments may be required. These will focus on identifying the best model for the preprocessing step detailed in Section 2.3.2. The goal is to achieve a performance comparable to the current preprocessing task, which boasts an accuracy of approximately 0.92.

**Author Contributions:** Conceptualization, Yizhou Tan; methodology, Yizhou Tan, Wenjing Li and Da Chen; software, Yizhou Tan and Da Chen; validation, Yizhou Tan; formal analysis, Yizhou Tan and Wenjing Li; investigation, Yizhou Tan; resources, Yizhou Tan and Wenjing Li; data curation, Yizhou Tan; writing—original draft preparation, Yizhou Tan; writing—review & editing, Yizhou Tan, Wenjing Li and Da Chen; visualization, Yizhou Tan; supervision, Waishan Qiu; project administration, Waishan Qiu. All authors have read and agreed to the published version of the manuscript.

**Funding:** This research received no external funding.

**Data Availability Statement:** Publicly available datasets were analyzed in this study. This data can be found here: https://data.world/city-of-ny/6eti-k994 (accessed on 9 December 2021).

**Conflicts of Interest:** The authors declare no conflict of interest.

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
