# Peer review of "Identifying Urban Park Events through Computer Vision-Assisted Categorization of Publicly-Available Imagery"

_ijgi, doi:10.3390/ijgi12100419_

Round 1
Reviewer 1 Report
This is a highly original and methodologically advanced paper that provides a novel analysis of park events staged in NYC parks. Strengths of the paper include some excellent visuals; especially the geographical distribution of event types. I am not really in a position to judge the robustness of the techniques used to analyse the imagery; as this is something I not familiar with. However, I can judge the significance of the paper in terms of its aims, outcomes and in terms of the issues it addresses. My sense is that the aims of the paper are potentially useful – but the purpose of the paper is not very well expressed at present. We need a much stronger case in the introductory sections for why this research is needed and what purposes the findings might serve. Similarly, at the end of the paper, the authors need to explain clearly how their findings can be used by ‘city planners and park administrators’ as at present this is account is rather vague and speculative. One aspect of the paper which is problematic is the categorization of events (pp.5-6). This seems fundamental, but the authors don’t explain in detail how they arrived at their 12 categories. There seems to be two main issues here: a degree of overlap between categories (e.g. Historic House Festival is in ‘festivals’ not ‘history’) and a mismatch between the scope of categories (e.g. GreenThumb which seems very narrowly defined and ‘family’ which seems very wide ranging). The definition of these categories exerts a strong influence over the findings – perhaps rendering them less insightful. Overall, I get the sense that the authors are expert in these techniques, but not particularly familiar with the parks / events fields. Therefore, the value of this paper seems to be in the methods used, rather than in the specific application here. The authors do a good job of explaining the potential of this type of research - and possible future extensions – but the outcomes and significance of the specific study outlined here need to be explained and justified in more detail.
Generally, clear (although note some erroneous punctuation marks on my pdf - maybe some glitch?)
Author Response
Dear reviewer, thanks a lot, we've learned a lot and enjoyed the revision process. Please see attached document for details.

Reviewer 2 Report
This manuscript trained CNN models to identify urban park events from images. Then some classical models were compared. I don’t think this manuscript is suitable for publication in this journal.
This manuscript focuses on CNN models for image classification. It does not conduct any work on spatial information. I don’t think its subject fits the aims and scopes of this journal. Journals in image processing or artificial intelligence would be more appropriate.
Author Response
Dear reviewer, thanks a lot, we've largely revised the manuscript accordingly. Really enjoy the revision process and have learnt a lot. Please see attached document for details.

Reviewer 3 Report
Thanks for submitting an interesting and impactful study. My overall critique is as follows:
Abstract
· One would expect to clearly see the main objective of this paper. It is therefore suggested to clearly mention objective of the paper.
1. Introduction
· Line 42: Identifying research gap with the help of literature review is acknowledged and appreciated
2. Dataset and Methods
· Line 155: The statement, “ To more efficiently identify events in urban parks, this research applies 154 Convolutional Neural Networks (CNNs) on images…..”
o What is the spatial, spectral and temporal resolution of the imagery? Please add it.
· Line 156: “We conduct data preprocessing with transfer learning….”. You can write summary of section 2.3 in two or three lines
· Line 158: “Secondly, we compare across different machine learning models”. Which models? Please name these different machine learning models.
· Line 161: Rephrase caption of Figure 1. For example, Overall research methodology
· Figure 2 and its caption is confusing. Please rectify and also mention the address of the website as data source.
· Line 186: “We manually grouped these categories into larger groups and formed 12 new categories”.
o What is the rationale for doing so? Please explain for clarity.
o What was the criteria to combine categories?
3. Results
· Line 295: Figure 4: Information in different colours refer to what? Please add legend /key to make Figure 4 understandable
4. Conclusion
· Line 389: Please rectify.
References need formatting. Please do that.
Moderate editing is required
Author Response

(The authors gave the same response as above.)

Reviewer 4 Report
The article is very interesting and I really enjoy reading it. I just have a few small annotations that can improve the quality of the article.
Starting the Introduction with a subsection “1.1. Background”. I understand the decision behind it, so this is only a suggestion, but personally, I would just start 1.1 on related works and leave the first paragraphs of the introduction without subsection.
The first phrase of the introduction can be improved in terms of vocabulary. “… has continuously evolved in the lives…” does not sounds that great. My suggestion is to find other words to pass the same idea.
LINE 37: “… mentioned…” might not be the best word for it. Perhaps analyse or took into consideration. This would required to rewrite the sentence.
LINE 42-45: Please add references of articles that perform said analysis. If possible, at least one for each type of analysis (intensity of park use, demographics, periods, and so on…).
LINE 46: Period before starting the new sentence.
LINE 46-59: The Secondly and thirdly is a bit muddy and the paragraph is confusing to read. When you write secondly you are referring to “the majorities of studies….”. The phrase that starts with “Recent technological…” changes the subject a bit and does not feel within the “secondly” mentioned before. From that moment forward it is written that GPS-based mobile phone is not informative and so is public participation, but the next sentence is not a negative, which leaves an awkward connection between two negatives and one positive. And then, comes the thirdly. So, in sum, I would try to rewrite this paragraph to make more sense to the reader and provide a better connection between subjects.
LINE 58-59 can connect to next paragraph. The current research status calls for an update… “In this article…” and continue to write the last paragraph, emphasising the importance, novelty and relevance of your paper.
LINE 135-151: This is also just a suggestion but perhaps don’t write in bullet points? I understand that the idea is to highlight the 3 research gaps and now the study address them, but maybe in just plain text plus a small table relating the gap with the work made in this article. This allows the reader to directly see in one column what is missing and right in front how this article address it.
Figure 1 needs to be a bigger to improve readability
Figure 4 and A1: As much as I appreciate the simplicity of the legend, it is incomplete. First, having the highest and lowest number is far from enough; considering there are several colours I would advise adding said colours to the scale. Talking of scale, the maps need a scale, need a north arrow and since I’m assuming that roads are also represented, it could also use a legend in that remark.
Additionally, before presenting the maps with the distribution of select parks events, it would be ideal to present of map of representing the location of the parks, so anyone can easily understand what parks are under evaluation and their location.
LINE 296-310: Some spelling and typing mistakes. They continue in the next paragraphs.
Parks characteristics and their impact on certain events is argued in the article but there are some characteristics missing that might also play an important role, such as the size of the park, the surrounding area (might be an area with higher art and music culture, or sports…), the vegetation and open space, orography… I would advice the author’s to take a moment and think of different park characteristics and how can they influence the events that can be held. This would make the article much more compelling and complete.
Moderate editing of English language required
Author Response

(The authors gave the same response as above.)

Round 2
Reviewer 1 Report
The authors have made a concerted effort to respond to my comments and made appropriate amendments. The paper has been improved. However, before publication I would like to see the conclusions improved further - in particular the authors need to suggest how planners/park authorities can use the findings of this study to make decisions re: how to 'allocate resources events, schedule events or develop infrastructure'. At the moment these implications are still very vaguely defined - I understand the auhors are trying to show the potential of this method, but it would be even better of they could show how this type of analysis might lead to specific recommendations.
Author Response
Reviewer #1:
The authors have made a concerted effort to respond to my comments and made appropriate amendments. The paper has been improved. However, before publication I would like to see the conclusions improved further -
1) In particular the authors need to suggest how planners/park authorities can use the findings of this study to make decisions re: how to 'allocate resources events, schedule events or develop infrastructure'. At the moment these implications are still very vaguely defined - I understand the authors are trying to show the potential of this method, but it would be even better if they could show how this type of analysis might lead to specific recommendations.
A: We appreciate your valuable feedback and understand the importance of demonstrating the practical implications of our study for planners and park authorities. While the primary objective of our paper is to introduce an efficient method for identifying park activities—addressing a long-standing challenge faced by scholars—we recognize the need to further elucidate how our findings can directly inform urban planning decisions.
To this end, we've enhanced our conclusion with the following paragraph to better articulate the potential applications of our research:
“This study holds notable significance, particularly in enhancing research and planning related to urban park use. While scholars have demonstrated a sustained interest in exploring urban park use—investigating aspects from health effects [9,10] to determinants of park use [8,14,18–20]—their methods have often been constrained by the inefficiencies of direct observation [15,17,32,34], interviews [18], and mass questionnaires [9,18,31]. The Convolutional Neural Network (CNN) method proposed herein provides a promising alternative, capably recognizing and distinguishing a wide array of activities within urban parks, thereby equipping researchers with a potent tool for future investigations. Furthermore, accurate categorization can aid city planners and park administrators in making informed decisions about resource allocation, event scheduling, and infrastructure development tailored to the unique needs of different event types. This is applicable in both the design and operational phases of park management. For instance, accurate event categorization can unveil the mismatches between existing park facilities and the prevalent types of events. To illustrate, the Union Square Park is a relatively compact space nestled between Park Avenue and 5th Avenue in New York City. Despite the absence of any sports-related facilities, ‘Sports’ emerged as the most predominant event type in the park, with 72 events labeled sports-related, many of which are fitness classes. This is notably higher than in other parks on the periphery of New York City, which are equipped with diverse sports facilities like tennis and basketball courts. Factors such as location convenience and community preference might have influenced this trend. Consequently, planners may contemplate integrating more facilities to accommodate fitness activities in Union Square Park. Additionally, such insights can shape the design of future parks, ensuring they are aptly equipped to support desired activities.”

Reviewer 2 Report
This manuscript is well written, but I still think it is not very relevant to the field of geographical information. Nonetheless, I suggest that the authors address the following two issues before the manuscript can be accepted.
(1) The authors should clarify how the location of an image is determined. For example, in figure 2, the location is “at Margaret Corbin Circle (in Fort Tryon Park)”. Is Margaret Corbin Circle or Fort Tryon Park chosen as the location of this image? This problem is also relevant to figure 5 and figure A1. What are the locations of the colored circles in these two figures?
(2) I suggest moving figure A1 into figure 5 and discuss the spatial distribution of each category of events in more detail.
Author Response
Reviewer #2:
This manuscript is well written, but I still think it is not very relevant to the field of geographical information. Nonetheless, I suggest that the authors address the following two issues before the manuscript can be accepted.
1) The authors should clarify how the location of an image is determined. For example, in figure 2, the location is “at Margaret Corbin Circle (in Fort Tryon Park)”. Is Margaret Corbin Circle or Fort Tryon Park chosen as the location of this image? This problem is also relevant to figure 5 and figure A1. What are the locations of the colored circles in these two figures?
A: Thank you for bringing attention to that. We have added the following sentences to clarify:
“For a visual representation of this distribution in Figure 5, we aggregated all events situated within the boundaries of each park. Then, circles are positioned at the centroid of each park's geometry or, if a park has multiple geometries, the centroid of the aggregated geometries associated with that park. The number of events for each category is indicated by both the color and the size of these circles.”
2) I suggest moving figure A1 into figure 5 and discuss the spatial distribution of each category of events in more detail.
A: Thank you for the suggestion! We have moved all figures in A1 into Figure 5, and discussed the distribution in more detail in the paragraph before Figure 5:
“Events categorized under ‘Film’ are prevalent across numerous locations, suggesting that many of these parks are equipped for outdoor film screenings or theatrical performances. The event categories of 'Family', 'Festivals', 'Games', and 'Community' also demonstrate a relatively uniform distribution across parks. Whether it's many parks hosting a high volume of such events or a minimal variance between parks with the most and least of these events, such distribution suggests that these activities generally necessitate less specialized infrastructure or equipment. Conversely, while the ‘Art’ category displays a peak value of 249 events at a single park, such events are less widespread. This limited distribution indicates that specialized facilities are needed for art events, possibly making them less accessible to residents citywide. Similarly, parks housing ‘History & Culture’ events, apart from the notable Central Park, predominantly include history-centric venues such as Roger Morris Park and Alice Austen Park. Parks featuring ‘Nature’ events are predominantly located towards the city’s outskirts, a placement that seems intuitive given the larger, more natural landscapes in those regions. ‘GreenThumb’ and ‘Volunteering’ events exhibit a more selective distribution, with a handful of parks like Windmill Community Garden for the former, and iconic spots like Central Park and Prospect Park for the latter, emerging as predominant hosts. ‘Education’ events are also highly concentrated in specific parks, namely Central Park, Wave Hill Public Garden, and Cultural Center, as well as Conference House Park. ‘Sports’, on the other hand, presents an intriguing pattern; while certain parks are hotspots, the distribution seems less governed by the presence of sports facilities and more influenced by factors like local community culture, park location, and proximity to organizations that might host fitness classes and related activities.”
